# Rearing of Bitterling (*Rhodeus amarus*) Larvae and Fry under Controlled Conditions for the Restitution of Endangered Populations

**DOI:** 10.3390/ani11123534

**Published:** 2021-12-11

**Authors:** Roman Kujawa, Przemysław Piech

**Affiliations:** Department of Ichthyology and Aquaculture, Faculty of Animal Bioengineering, University of Warmia and Mazury in Olsztyn, 10-719 Olsztyn, Poland; reofish@uwm.edu.pl

**Keywords:** bitterling, controlled conditions, fry, larvae, rearing

## Abstract

**Simple Summary:**

Among the many species threatened with extinction and covered by protection of species is bitterling Rhodeus amarus. It belongs to ostracophilic fish that place spawn inside live mussels. Bitterlings, such as mussels, belongs to indicator species that testify to the good state of the natural environment. Supporting the populations of these organisms is a necessity in order to preserve the biodiversity of inland waters, which are subjected to severe anthropopression. The development in advance of a method of reproduction and breeding of bitterling under controlled conditions can ensure their survival in the event of an imbalance in the natural environment. These were the first studies of this type, where after 6.5 months of rearing, sexually mature individuals were obtained. In this way, a suitable stocking material of the bitterling was obtained in order to save the endangered populations.

**Abstract:**

Among the several dozen European freshwater fish species, only European bitterling (*Rhodeus amarus* Bloch) and *Rhodeus meridionalis* belong to the group of ostrakophilous fish. The embryonic and larval development of the fish in this reproductive group until the time of the yolk sac resorption takes place in the gill cavity of river mussels (*Anodonta* sp. or *Unio* sp.). This paper presents the results of the European bitterling *Rhodeus amarus* being reared under controlled conditions. Bitterling larvae were caught together with river mussels in the natural environment and subsequently placed in a tank for behavioural observations. Bitterling larvae were seen swimming in the water within a week of placing the bivalves under controlled conditions. The bitterling larvae were 8.6 ± 0.11 mm long when they started to swim actively. The rearing was conducted in water at 20 and 26 ± 0.5 °C and lasted for 6.5 months (200 days) in both variants. Initially, the larvae were fed with live nauplii of *Artemia salina* and subsequently with fodder. The bitterlings in tanks with water at 26 ± 0.5 °C were 66.2 ± 3.0 mm long and weighed 3389 ± 548 mg. For comparison, bitterlings kept in water at 20 ± 0.5 °C were 64.48 ± 3.4 mm long and weighed 3242 ± 427 mg. No larval malformities or mortality were observed during the larvae and fry rearing. The bitterlings had well-developed secondary sexual characteristics and exhibited pre-spawning behaviour at the end of the rearing. This produced suitable bitterling stocking material to be used in the conservation of small or endangered populations.

## 1. Introduction

The majority of over two dozen species of the genus Rhodeus described in the literature occur in Asia. There are only two bitterling species living in Europe: *Rhodeus meridionalis* found only in the Balkans, and *Rhodeus amarus*. *Rhodeus amarus*, inhabiting the majority of freshwaters of Europe, including Poland [1,2,3,4]. Currently, the bitterling is regarded as a native species in a large part of its distribution range in Europe. However, it had only occurred in the Ponto-Caspian region before the year 1100. The earliest mention of the bitterling in Western and Central Europe come from regions where carp breeding was common and bitterling spread along with the gradual carp breeding expansion. After the initial expansion period, the species practically disappeared from Europe during the coldest part of the Little Ice Age. Bitterlings reappeared in the late 18th century. Their presence was recorded in historical carp breeding centres. The bitterling became a common species in Europe after the mid-19th century. It did not reach the current range until the 20th century. This was associated with intensive bivalve expansion, temperature increase and anthropogenic changes in the environment, which all favoured bitterlings. Bitterlings are found most frequently in the littoral zone of lakes, shallow oxbow lakes, and in slow-flowing rivers, where they can be found in groups of several or even several hundred individuals. However, their presence depends on the occurrence of mussels: *Unio* sp. and *Anodonta* sp. [5,6,7]. Bivalves are indispensable for bitterling reproduction because the fish spawn can be incubated only in the gill cavities of mussels [8,9,10,11,12]. Bitterlings become sexually mature in second year of life under natural conditions [2,6]. One female bitterling with a mean length of 5–6 cm can lay several dozen to several hundred grains of large, non-sticky spawn into the gill cavities of several mussels. The number of eggs is closely related to the female size [7,12]. Even several dozen eggs can be laid into one mussel. According to literature reports, as many as 200 bitterling embryos can be found in one bivalve [9]. The water temperature ranges from 17 °C to 26 °C during the spawning period. Bitterlings develop inside a bivalve after fertilization. Larvae swim out of their incubators after 3–4 weeks (depending on the water temperature) band form shoals of several to several hundred larvae very soon thereafter [13,14]. They prefer shallow places, with dense vegetation, where they are not exposed to attacks of predators, i.e., other fish and aquatic insects, larvae and adults, such as great diving beetle *Dytiscus marginalis*, the beetle *Cybister lateralimarginalis* [15], or dragonfly larvae *Anax* sp. of the family *Aeshnidae* [16]. The bitterling population size dropped in 1960–1980 in Western and Central Europe, mainly due to water pollution. The decrease was so significant that strict national and international regulations aimed at the species conservation were implemented. The bitterling was entered on the list of endangered species and is now covered by strict all-year protection [17]. Currently, the bitterling existence in Polish waters is not endangered, but the species is protected all the same. This is due to the unsatisfactory status of bitterling populations in other European countries. A decrease in the bivalve populations (their natural incubators) is the greatest threat to bitterlings. Bivalves are susceptible to natural aquatic environment pollution, and since they live a sedentary lifestyle, they cannot “swim away” from polluted places.

The current study aimed to examine the feasibility of prolonged rearing of bitterling larvae and fry under various thermal conditions. It was assumed that bitterling spawners ready for reproduction can be obtained after a 6.5-month-long rearing period. The objective of the study was to develop a bitterling rearing technology aimed at the restitution of endangered bitterling populations. The bitterling larvae rearing technology under controlled conditions will support the survival of local populations [7,18], whose size has rapidly declined as a result of poisoning or other adverse environmental changes.

## 2. Materials and Methods

Bitterling larvae transferred in *Unio* spp. mussels from a natural water body were used in the study. The mussels were obtained several days after the bitterling spawning was observed. The lake Bartążek (Poland) water temperature was 22 °C at spring. 23 mussels were transported in tanks with fresh water. Subsequently, they were placed in a tank with water of the same temperature for behavioural observations. The water temperature in the aquarium was increased to 24 ± 0.5 °C over 12 h. The tank with mussels had a several-centimetre-thick layer of gravel and sand, and the water was slightly aerated and filtered. The experiment was conducted at the Laboratory of the Department of Ichthyology and Aquaculture in the Centre for Aquaculture and Ecological Engineering of the University of Warmia and Mazury in Olsztyn. The first bitterling larvae were observed within a week of placing the bivalves under controlled conditions. All the bitterlings left their incubators, filled their swim bladder with air and swam in search of food within three days. The bivalves were transported back to the lake from which they had been brought. The breeding of the bitterling larvae was carried out in two stages.

In the first stage, bitterling larvae were placed in six flow-through aquariums with a capacity of 65 dm^3^, but initially, for 1.5 months, they were only partly filled with water so that the working volume was 20 dm^3^ each. The aquaria were placed in two water recirculation systems with water temperatures of 20 °C and 26 ± 0.5 °C. Two hundred bitterling larvae were placed in each tank. Each of the recirculation systems had effective water filtration (an external EHEIM 2280 filter) with maximum capacity of 1.200 L h^−1^. The water flow rate through the aquaria was 0.25 L·min^−1^. The aquaria were lit with 54 W of light from 8.00 a.m. to 8.00. p.m. (LD12:12). In the second stage, after a 1.5-month period of raising the bitterling larvae, the water volume in the storage tanks was increased from 20 to 60 dm^3^ each. This was achieved by lifting the drain. Breeding in these conditions was carried out for 5 months. Live nauplii of *Artemia* sp., prepared in accordance with the formula described by Sorgeloos [19], were fed to the larvae. The nauplii of Artemia sp. (stage I nauplii) were 430 μm long. After a week, the larvae started to receive small amounts of Nutra HP 0.5 fodder, manufactured by Skretting. After three weeks of rearing, the larvae received only fodder comprised of specialist fry feed with granulation of 0.5 mm. The feed, both Artemia spp. nauplii and the fodder, was given manually three times daily, every four hours. The fodder doses were adapted to the weight of the fish being reared and adjusted after a sampling day. The composition of the natural feed and the fodder used in the experiment is shown in Table 1. The rearing lasted for six months. The oxygen content of the water was measured with a YSI multi-parameter meter. The ammonia and nitrate content was measured with a HANNA INS HI 83200 with reagents. The oxygen content was within 7.2–8.2 mg O2·dm^−3^; the water pH varied from 7.2 to 7.3. No ammonia or nitrites were detected in the water during the experiment.

### Sample Collection and Preparation

A control sample of larvae (30 larvae) was collected on the day the aquaria were stocked and before feeding was started. Subsequent samples (20 larvae from each aquarium) were collected every 6–8 days in the first stage of research and every 30 days in the second stage of research. Samples were always collected before feeding was started. Before being measured, the larvae were anaesthetized in a solution of MS-222 (tricaine methanesulfonate) (20–50 mg·L^−1^ of water). After being caught, the larvae were weighed on a KERN ALJ 220-5 DNM analytical balance. Subsequently, the anaesthetized fish were placed on a Petri dish and observed under a Leica MZ16Z stereoscopic microscope. The photographic documentation of the fish was made with a DFC 420 microscopic camera. Their total body length (longitudo totalis—l. t.) was measured at an accuracy of 0.1 mm. The fish size was analysed with LAS software V 3.1.0. After the measurements, the larvae were returned to their corresponding tanks.

The measurements were used to calculate the relative body growth rate (SGR), the index of increases in the total length per time unit (ITL), physical condition, survivability, and larvae biomass. The relative specific body growth rate (SGR) and relative biomass growth rate (SBR) from the onset of feeding until the experiment was terminated were computed from the formulas [20]:$$\mathrm{SGR}=100\;\cdot \;\frac{\ln W_{2}-lnW_{1}}{\Delta t},\;\;\mathrm{and}$$
$$\mathrm{SBR}=100\;\cdot \;\frac{\ln\left(n_{2}\cdot W_{2}\right)-ln\left(n_{1}\;\cdot W_{1}\right)}{\Delta t}$$
where: 

W_1_—mean initial weight of a reared individual (mg);

W_2_—mean final weight of a reared individual (mg);

n_1_—number of individuals (indiv.) at the onset of rearing;

n_2_—number of individuals (indiv.) at the end of rearing;

t—duration of the rearing-up period (days).

Next, based on the above, the relative rate of body gains (RGR) and relative rate of the biomass increase (RBR) from the onset of feeding to the termination of the experiment were computed from the following formulas [21]:$$\mathrm{RGR}=100\;\cdot \;\left(e^{\frac{SGR}{100}}-1\right),\;\mathrm{and}$$
$$\mathrm{RBR}=100\;\cdot \;\left(e^{\frac{SBR}{100}}-1\right)$$

The rates of increments (the SGR and RGR) for the body length were calculated analogously. The increase in the total length (the ITL) per time unit [mm/d] was derived from the formula [22]:$$\mathrm{ITL}=\frac{TL\;\left(n_{2}\right)-TL\;\left(n_{1}\right)}{\Delta t}$$
where:

TL—mean length of an individual (longitudo totalis);

n_1_—start of the period;

n_2_—end of the period;

t—duration of the rearing (days-d).

The physical condition of larvae was computed from the Fulton’s equation Nielsen and Johnson:K = 100 WL − 3
where:

W—mean weigh of an individual (mg);

L—TL (mm).

The biomass of fish in each tank was determined by multiplying the mean individual weight and the number of live individuals. The value thus obtained was divided by the capacity of the tanks. The biomass computed as above was expressed in g·dm^−3^. The statistical differences between the groups were demonstrated with the Duncan’s test (1955) at the level of significance of a = 0.05. The statistical analysis of the results was performed with Microsoft Excel 2019 and Statistica 12.0 for Windows.

## 3. Results

Bitterling larvae at the onset of active swimming were 8.6 mm (±0.1) long on average. During the first stage of rearing, which lasted for 45 days, very good results were obtained, in terms of both the growth and survivability of bitterling larvae (Figure 1 and Figure 2).

From day 7 of the rearing on, both mean body weights and mean body lengths of larvae kept in the water of 20 °C in temperature were statistically significantly different from those of the larvae maintained in the warmer water. From day 22 of the rearing, the differences in the mean body length between the individuals reared in the water of the temperature 20 °C and 26 °C remained basically on the same level, whereas differences in the body weight between these two groups increased exponentially. As soon as four weeks into the rearing, juvenile bitterlings kept in the warmer water resembled adult specimens in body shape. As the end of the first stage, bitterlings reared in the water of the temperature 26 °C reached the average body weight of 720.8 mg (±80.4) and the average body length of 40.2 mg (±2.1). Meanwhile, those kept in the warmer water, of the temperature of 20 °C, reached 298.5 mg (±32.2) of average weight and 34.4 mg (±1.67) of average length at the end of the rearing period. This meant they weighed nearly 2.5-fold less than their peers reared in the warmer water. Likewise, their rearing parameters, such as ITL, RGR, or the Fulton’s index, calculated on the basis of the measured body weights and lengths, were lower than the ones achieved for the bitterlings reared in the water of the temperature of 26 °C (Table 2). The increase in the total body length per unit of time (ITL) calculated for the bitterling fry from the water of the temperature of 20 °C and 26°C was 0.57 and 0.7 mm·d^−1^, respectively. The biomass of the fish reared in the warmer water at the end of the 1.5-month-long rearing period was 7.21 g·dm^−3^. Under the analysed thermal conditions, the RGR and RBR were higher for the fry maintained in the warmer water. During the second stage of rearing, which lasted for 150 days, the growth of bitterling fry in the water of the temperature of 26 °C was observed to gradually slow down. This was most probably caused by the fact that the fish achieved sexual maturity, which was manifested by the appearance of the mating robe. The growth of the bitterling fry in the colder water proceeded more rapidly, which is reflected by the computed values of rearing indices. At the end of the rearing period, bitterlings kept in the colder water decelerated their growth when reaching sexual maturity. Despite the initial differences in the body weight and length growth, no statistically significant differences were determined at the end of rearing between the fish originating from the two temperature regimes. At the end of the 6.5-month-long rearing period, the bitterlings from the water of the temperature of 20 °C reached the average body weight of 3242 mg (±427) at the average body length of 64.48 mm (±3.4). In the warmer water, they reached the average body weight of 3389 mg (±548) and the average body length of 66.2 mm (±3.0) (Figure 3 and Figure 4).

The values of the rearing indices calculated at the end of the rearing period were approximately the same for both water temperature regimes. At that time, the body length increase per unit of time (ITL) calculated for the bitterling fry kept in water at 20 °C and 26 °C was 0.20 and 0.17 mm·d^−1^, respectively. The biomass of fish reared in the warmer water at the end of an over six-month rearing period was 11.3 g·dm^−3^. The values of such indicators as the RGR and RBR were higher for the bitterling reared in the cooler water. Throughout the entire rearing period, no fish losses were recorded in either temperature regime, despite the fact that the fry were periodically subjected to manipulations when taking the measurements of their body weight and length. Based on the results of these measurements during the first and second stage of the rearing period, a curve was plotted illustrating the dependence between the length and the weight of the fry. This curve is shown in Figure 5.

## 4. Discussion

The method for rearing up bitterling larvae and then fry perfectly responds to the research current aiming to develop methods for reproduction and rearing of fish species threatened with extinction. The development of biotechniques enabling the reproduction and rearing of threatened species allows us to take measures early enough to prevent their extinction. Some examples are studies dedicated to the reproduction and rearing of such species as: spined loach *Cobitis taenia* [23,24], European weather loach *Misgurnus fossilis* [25], lake minnow *Rhynchocypris percnurus* [26,27], and river lamprey *Lampetra fluviatilis* [28,29]. The available literature lacks reports on research dealing with the rearing of bitterling larvae and then fry under controlled conditions, and therefore the results presented below should fill in this gap.

Restitution of populations of bitterling, a fish species threatened with extinction, can be achieved in several ways. One is to transfer mussels and adult bitterling fish from other water bodies. Another one is to transfer only mussels from water bodies in which bitterling were observed to show spawning behaviour. It is also possible to reproduce bitterling under controlled conditions, but this is rather difficult due to the small size of spawners and the presence of an ovipositor in females. Much experience, diligence, and gentle manipulation are needed to obtain roe and semen from bitterling spawners under controlled conditions. Moreover, the method is inefficient and time-consuming because bitterling release spawn in batches. Once fertilised, the spawn can be incubated in specially designed incubators [7].

Following many years of research, it has been concluded that the most effective way of obtaining bitterling fry for fish stocking is a combined method, where mussels are collected from their natural habitats during the spawning season of bitterling. Larvae which hatch from the eggs deposited inside mussels will remain inside the mussel for one up to two weeks, depending on the water temperature. Afterwards, fully formed larvae will swim out in order to search for exogenous food. This is the right moment to start their rearing under controlled conditions. It is worth bearing in mind that small and medium-sized specimens are recommended when collecting mussels for this purpose. Direct observations prove that bitterling prefer smaller mussels for depositing eggs [6]. Large ones tend to be ignored because they are able to remove the deposited eggs [10,11,12,30,31,32]. A strong contraction of the mantle muscle helps a mussel to remove bitterling embryos [33].

The same tanks were used for rearing up the bitterling larvae and then fry, although in the early stage (45 days) they contained less water, equal the working capacity of 20 dm^3^. After 45 days of rearing, the water table was raised to ensure the working capacity of 60 dm^3^. This is a very practical solution because it ensures suitable conditions for both larvae and fry (better access to food, less stress). Using the same tanks where the water height is adjustable is a solution that can be employed in the long-term rearing up of larvae and then fry of other fish species, e.g., sampa *Heterobranchus longifilis* [34].

This study revealed varied rates of growth of bitterling larvae and fry maintained in water of different temperature regimes: 20 °C and 26 °C. It was observed that when bitterling larvae and subsequently fry were reared, the larvae in the warmer water grew more rapidly at first but their growth rate slowed down in a few weeks. In contrast, the larvae and then fry kept in the cooler water grew more slowly but at the end of the rearing period achieved the body size similar to the one obtained by fish in the warmer water. It is difficult to explain what causes such a growth pattern and whether this is the rule that fish from colder water reach a size similar to that obtained by fish from warmer water during a long-term rearing. It may depend on the rate of absorbing food. Initially, live nauplii of Artemia sp., and later commercial feed were fed to bitterling larvae. The literature data indicate that young specimens, with the body length to 40 mm, feed on both plant and animal food [35], whereas larger individuals prefer low-digestible food, such as detritus or plant residue, while the share of invertebrates decreases with an increasing body length of the fish [36]. While taking control measurements during the entire rearing period, no body deformations were observed on larvae and fry from both lower and higher temperature water that might have been caused by an inappropriate diet. The fact that food could be a cause of body deformation among fish larvae and fry has been reported by Kamiński [37], based on their research on rearing lake minnow.

During the rearing period, compensatory growth may have occurred, which enabled the bitterling larvae and then fry maintained in the colder water to achieve a similar body size. In contrast, bitterling reared in the warmer water might have experienced (a period of) the flattening of the growth (curve), when their growth was decelerated. It is probable that larvae kept in the colder water, after some rearing time, went into compensatory growth, which let them compensate for the growth possibilities lost in the earlier stage. The mechanism of compensatory growth in different fish species has been described by many authors Xie—gibel carp *Carassius auratus gibelio* [38], Bélanger—Atlantic cod *Gadus morhua* [39], Nikki—rainbow trout *Oncorhynchus mykiss* [40], Van Dijk—roach *Rutilus rutilus* [41], and concerned mostly the fish whose growth rate was initially slower due to a change in feed or limited food availability. Hyperphagia could be the mechanism underlying the accelerated growth of undernourished fish during compensatory growth described in the mentioned papers. Changes in the growth rate following a period of some shortage of feed have also been described in the case of tilapia *Oreochromis mossambicus* × *O. niloticus* by Wang [42]. Relative protein, lipid, and ash gains, as proportions of the total body mass gain, did not differ significantly between the analysed groups of tilapia. However, it is still difficult to identify precisely the mechanism responsible for the levelling of the growth results between the two groups of bitterling. This problem requires a series of research and comparative studies.

The range of temperatures used during the rearing of larvae and fry of bitterling described in this manuscript agrees with the one recommended during the rearing of other fish species reproducing in a similar season e.g., lake minnow *Rhynchocypris percnurus* [26,27] and sichel *Pelecus cultratus* [43].

The above study showed that bitterling larvae, same as larvae of other fish, can be reared up quite successfully under controlled conditions. When provided with optimal feeding and thermal conditions, the fry will achieve the body weight of 300 mg and length of 30 mm after 30 days of rearing. This is half the size of bitterling when they obtain sexual maturity. A rearing period lasting for 6.5 month in total, both in the warmer and colder temperature, enabled us to produce sexually mature fish, evidenced by the males exhibiting nuptial colour and pearl rash [2,7,9].

The size of fish, coloration of males, occurrence of pearl rash and characteristic pre-spawning behaviour proved that the fish reached sexual maturity and were ready for spawning [5]. This conclusion was supported by years of observations of the behaviour of bitterling during spawning in both natural and controlled conditions. It was further confirmed when (after the experiment described herein was terminated) selected bitterling specimens were placed in another aquarium, with mussels, where they began to spawn. Such behaviour of bitterling suggests that it is not necessary (as has been thought so far) to submit the fish to vernalisation, that is to keep them in water of lower temperature before spawning. They are stimulated by the presence of mussels and a suitable temperature of the water. Thus, it is possible to reproduce bitterling a few times a year and to use the reared-up material for stocking the water bodies chosen for the reintroduction of this species.

The method for obtaining bitterling larvae and their rearing up, presented in this manuscript, is extremely useful for reconstitution of endangered populations. When we acquire larvae from their natural habitat and rear them in controlled conditions, we guarantee high survivability during the most perilous stage of life. The method allows us to preserve the gene pool of parent individuals, which are perfectly adapted to the living conditions defined by the habitat in which they dwell [44,45,46]. The initial response of individuals from a given population to environmental changes is frequently behavioural, and it depends on the genetically conditioned response of individuals within this population, which has evolved through generations. Afterwards, certain genetic changes appear, which facilitate the population’s adaptation to new conditions [47]. Such fish stocking material possesses all genetically conditioned traits which improve survivability and reproduction success in a given environment [48,49].

However, for the reintroduction to be successful, first we need to ensure proper environmental conditions for mussels, which should precede bitterling fish in the chosen habitat. Mussels are indicator organisms, highly sensitive to water pollution. Hence, if they are observed to be dying, it is not worth introducing bitterling fry to this water body. First and foremost, the condition of this water habitat must be improved until it is suitable for mussels, and then bitterling can be reintroduced into this habitat. There is one more reason why the above measures can be successful. Bitterling are among the fish that are hardy and able to rebuild their population even after a severe decline in their number, provided the water body is abundant in mussels from the family *Unionidae*. The bitterling fish fry obtained from our experimental rearing of larvae and fry is an excellent fish stocking material to stock water bodies in which bitterling populations are at risk of becoming extinct, or to which wastewater has been discharged annihilating the entire population of these fish.

## 5. Conclusions

The above experimental study demonstrated that it is possible to rear the bitterling larvae and fry under controlled conditions. During the first stage of the experiment, the bitterling larvae reared for 45 days in the warmer water grew more rapidly. However, during the second stage of the study, when the bitterling fry was reared for 150 days, the rate at which their weight and body length increased was much lower. At the end of rearing, both in the cooler and warmer water, the bitterling fish obtained were ready for reproduction, which was confirmed by the appearance of the mating robe. Throughout the entire rearing period, no losses were observed among the fry, despite the fact that they were periodically submitted to manipulations in order to take measurements of their body weight and length. The bitterling specimens reared during the experiment (during a 6.5-month long rearing period) can be an excellent stocking material in sites in which, for various reasons, populations of this fish have diminished drastically.

## Figures and Tables

**Figure 1 animals-11-03534-f001:**
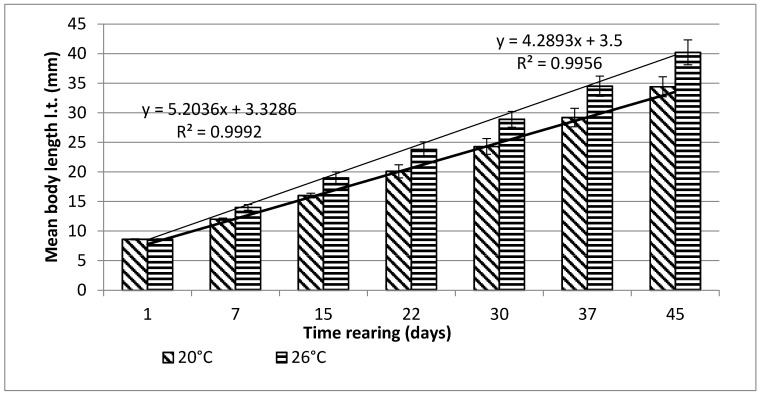
Stage I—Increase in the body length of bitterling larvae during rearing (45 days) in water at different temperatures. Data are shown as means ± SD.

**Figure 2 animals-11-03534-f002:**
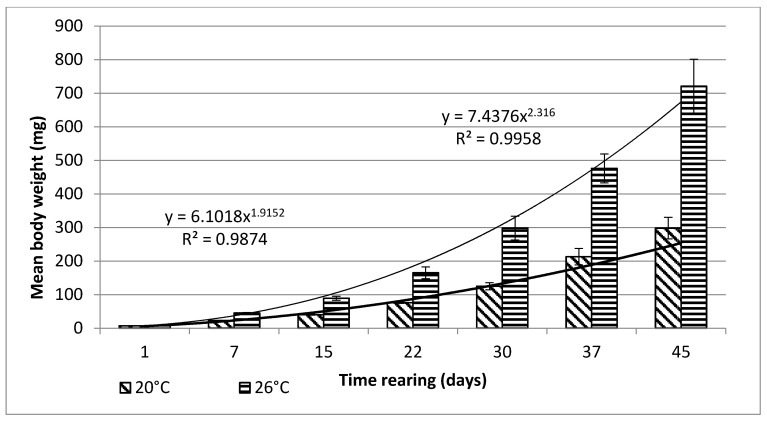
Stage I—Increase in the body weight of bitterling larvae during rearing (45 days) in water at different temperatures. Data are shown as means ± SD.

**Figure 3 animals-11-03534-f003:**
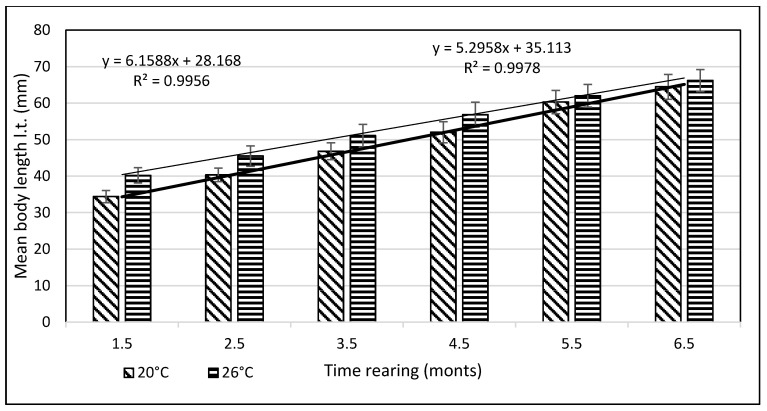
Stage II—Increase in the body length of bitterling fry during rearing (5 months) in water at different temperatures. Data are shown as means ± SD.

**Figure 4 animals-11-03534-f004:**
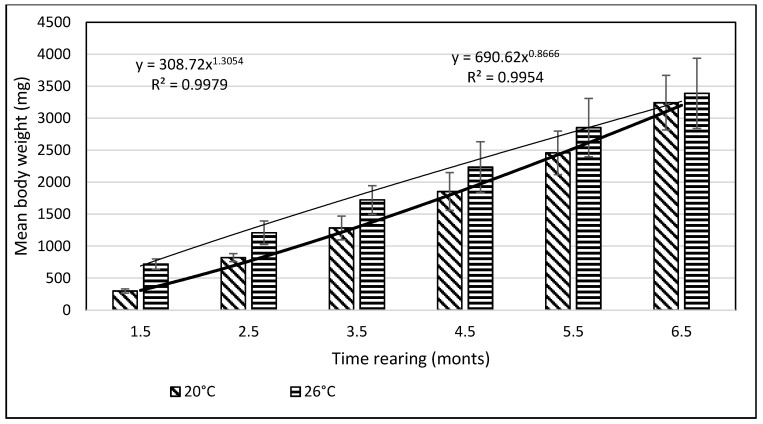
Stage II—Increase in the body weight of bitterling fry during rearing (5 months) in water at different temperatures. Data are shown as means ± SD.

**Figure 5 animals-11-03534-f005:**
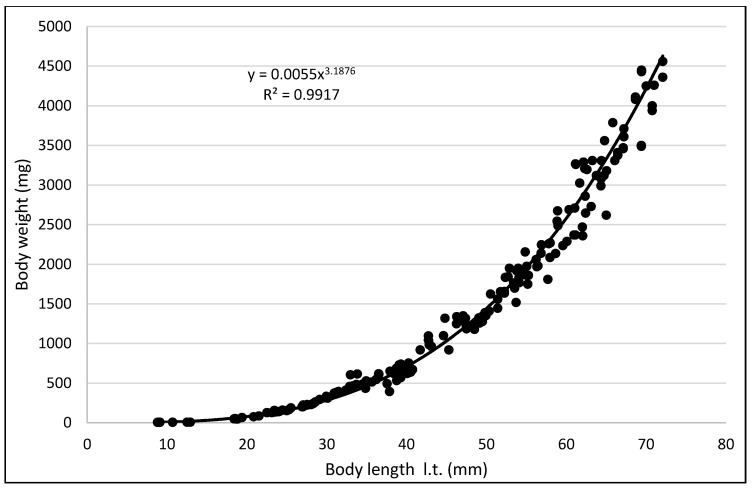
Relationship between length and weight of bitterling.

**Table 1 animals-11-03534-t001:** Chemical composition of Artemia nauplii * and fry fodder Nutra HP ** (% of dry weight) (*—data according to P. Candreva. Inve Aquaculture NV. Dendermonde. Belgium, ** Skretting).

Feed	Component (%)				
	Protein	Fat	Carbohydrates	Ash	Phosphorus
*Artemia* sp.	47.0	21.5	10.6	9.5	-
Nutra HP	55.0	18.0	0.5	10.5	1.7

**Table 2 animals-11-03534-t002:** The results of rearing bitterling *Rhodeus amarus* larvae and fry in two water temperature.

Parameter	I STAGE (Time Rearing 1.5 Months)		II STAGE (Time Rearing 5 Months)	
	Temperature (°C)		Temperature (°C)	
	20	26	20	26
Initial mean body weight (mg)	7 ± 0.23 ^a^	7 ± 0.24 ^a^	298 ± 45.8 ^a^	721 ± 97.55 ^b^
Final mean body weight (mg)	298 ± 32.2 ^a^	721 ± 80.4 ^b^	3242 ± 427 ^a^	3389 ± 548 ^a^
Initial mean body length (mm)	8.6 ± 0.11 ^a^	8.6 ± 0.12 ^a^	34.4 ± 0.63 ^a^	40.2 ± 1.98 ^b^
Final mean body length (mm)	34.4 ± 1.67 ^a^	40.2 ± 2.1 ^b^	64.48 ± 3.4 ^a^	66.2 ± 3.0 ^a^
Initial stock (indiv.)	200	200	200	200
Final stock (indiv.)	200	200	200	200
Survival (%)	100	100	100	100
Time rearing (days)	45	45	150	150
Increase in total lenght (ITL) (mm d^−1^)	0.57 ± 0.02 ^a^	0.70 ± 0.04 ^a^	0.20 ± 0.01 ^a^	0.17 ± 0.02 ^b^
Relative growth rate (RGR) for weight (% d^−1^)	8.69 ± 0.4 ^a^	10.85 ± 0.6 ^b^	1.60 ± 0.17 ^a^	1.03 ± 0.24 ^b^
Relative growth rate (RGR) for length (% d^−1^)	3.13 ± 0.1 ^a^	3.47 ± 0.2 ^b^	0.42 ± 0.02 ^a^	0.33 ± 0.04 ^b^
Relative growth rate (RBR) for biomass (% d^−1^)	8.69 ± 0.4 ^a^	10.85 ± 0.6 ^b^	1.60 ± 0.05 ^a^	1.03 ± 0.03 ^b^
Biomass (g dm^−3^)	2.98 ± 0.3 ^a^	7.21 ± 0.51 ^b^	10.81 ± 0.62 ^a^	11.3 ± 0.53 ^a^
Fultona K	0.73 ± 0.06 ^a^	1.11 ± 0.09 ^b^	1.21 ±0.3 ^a^	1.2 ± 0.2 ^b^

Mean value ± S.D. Results in rows from the same stage with the same index of letters are not statistically significantly different (*p* ≤ 0.05).

## Data Availability

The data presented in this study are available on request from the corresponding author.

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
