# Peer review of "Rearing of Bitterling (Rhodeus amarus) Larvae and Fry under Controlled Conditions for the Restitution of Endangered Populations"

_animals, 2021, doi:10.3390/ani11123534_

Round 1
Reviewer 1 Report
The work is innovative and the feeding methods used for such a
long period of time give a new tool in aquaculture.
In my opinion, the planning of the experiments and the way
of describing the results do not contain factual errors.
The authors' conclusions provide new opportunities in the breeding
of Rhodeus sericeus.
Author Response
Dear Reviewer,
Thank You very much for your review. All comments were taken during review process marked in the text. All main changes were marked in the text in blue.
We appreciate for Your time and revision.

Reviewer 2 Report
The paper presents interesting results of research on the European bitterling rearing under controlled conditions. In the history of research , this species had a different status in terms of its protection. At the same time, this fish is an interesting example of an organism used for reproduction some clams of the genus Unio or Anodonta. Changes made by man in the environment inhabited by bitterling (including water pollution) lead to a reduction in the number of its natural populations. This can lead to the displacement of the bitterling from many water bodies that have so far been inhabited by this fish. The authors present a method of obtaining bitterling larvae from the natural environment and their rearing in order to return the material to the environment from which it was collected. It is worth emphasizing that the method ensures obtaining fish in good condition, which is important in terms of their further use in natural environment.
The presented manuscript clearly describes all stages of the work as well as the obtained results and conclusions resulting from them. Nevertheless, I do have a few minor comments/suggestions that the authors may take into account when compiling a final version of the paper.
First, the authors use several different Latin names of the described species in their work. It seems to me redundant and rather makes the reception of the work more difficult. Fishbase gives the official species name as Rhodeus amarus (Bloch, 1782). Among the synonyms given in the above database there is Rhodeus sericeus (used by the authors of this manuscript in the title) or Rhodeus sericeus amarus (Introduction, line 30). Unfortunately, both of these names are listed as synonyms which have expired. I ask the authors to verify these names and standardize them throughout the manuscript.
Second, the works presented by the authors are an interesting tool supporting the restoration of the biological diversity of ichthyofauna. It seems to me that there is no emphasis here that the presented method allows for the introduction with the use of native material. It is extremely important from the point of view of genetics and ecology of any species including R. amarus. I suggest that the authors emphasize this very important aspect of the work as a great advantage (it does not involve the use of material from different populations). Certainly, such information should also be included in the Abstract or Conclusions.
Moreover, in the Materials and Methods section, the authors state that the larvae came from a natural reservoir but they do not provide further information on it - please complete the data.
In the same section, I propose to clearly highlight stages I and II, as in the present form of the description (a uniform and continuous style) it seems to be slightly hidden in the text. It is a pity as it is very important from the reader's point of view.
Line 114 of the same chapter - we find such a description: "LAS V 3.1.0. Software”. Shouldn't this part be as follows: “LAS software v. 3.1.0. "?
While the SBR formulae (line 122) looks strange in the PDF file I received - please verify its correctness.
In the Keywords section, I suggest using an alphabetical order.
Author Response

(The authors gave the same response as above.)

Reviewer 3 Report
This is an interesting study concerning conservation. The paper is generally well written and structured. However, in my opinion the paper has some shortcomings regarding the discussion, which is too short and vague. A lack of a discussion of compensatory growth effects between both groups. Below I made additional suggestions for more in-depth analyses of the data. In my opinion, the conclusions regarding the rearing of Bitterling larvae are not well supported by the data.
The focus presented in the discussion is in considerable proportion based on bivalves when the objective of this study was on fish. From line 208 to line 231 does not add much to the discussion; this part should be shorter if kept.
the authors do not mention anything about the compensatory growth effect observed in the group of fish reared at 20 degrees Celsius. Why?
How do the authors explain this? What is the meaning of this trade-off regarding the onset of reproduction?
In conclusion, the authors will be paid more attention to elucidating this in the discussion section to improve the article and novelty.
LINE 48, please add the scientific name to dragonflyes.
LINE 68, please be more specific in “that time”, ie. Winter, summer, etc?
LINE 68 Eliminate Several and parenthesis. 23 musssels...
LINE 69, fresh or marine water? Would you mind clarifying this?
Line 70-71. This is not accurate. Which is the reason for increasing temperature to 24? Add the Celcius degrees symbol.
LINE 86. Would you please use the standard nomenclature for photoperiod (LD12:12)?
LINE 122. SBR formula presents an error because an Asian symbol appears. Typo.
Figures 1 and 2. In both plots, the statical analysis should be described in the footnote. Does the error bars an average and SEM?
LINE 183. At the end of the rearing period, bitterlings kept in the colder water decelerated their 183 growth when reaching sexual maturity. In my opinion, this belongs to the discussion section.
Table 1. In my opinion, this table does not give more information, and I suggest removing it and incorporating this information in the material and method sections.
Given these shortcomings, the manuscript requires major revisions.
Author Response

(The authors gave the same response as above.)

Reviewer 4 Report
The manuscript with code is animals-14696 is about the rearing of an endangered fish species Rhodeus sericeus at two different temperatures. The soundness of the data are interesting, however the authors did not progress through the reproduction of this specie in captivity that would be a high innovation in the field.
The experimental design of the manuscript is well performed and the conclusions are accurate. However several minor points should be addressed by the authors.
As the authors write down they have 6 tanks with 200 larvae each at two different temperatures. So, does it mean that each temperature was performed in triplicate tanks? Please write this issue clearly in the manuscript.
The same occurs with the sampling. 20 larvae for each tanks means 60 larvae for each temperature. Please clarify.
Authors should include how they measure oxygen, and ammonia. Did the authors measure nitrate?
The international units for biomass is Kg m-3, as the fish are very small authors give this data in g dm-3 that it is right, however at line 144 the units are g L-3. Please check this unit and correct them when needed.
The authors should divide the results into two minor points: a point including the data of the first stage of rearing and the other one, including the second stage of rearing. It would clarify several issues and improve comprehension of the results. For example, when the fish reached maturity?
Authors should included the SEM or de SD of the initial data as they measured 30 larvae in the text.
Lines 160-162. It would be nice to know the fold change observed between fish at 20 and at 26 at each sampling point.
Line 164. It would be nice whether the authors might show any picture of the process of maturity and show the fish that resembled adult specimens in body shape compared with the fish at colder temperature.
Line 178. What this means? Is there a table 2 that was not included in the paper?
Line 175. What about the biomass of the colder group?
Line 184. Authors should show the features that induced them to claim that fish reached sexual maturity
Line 197. Why the authors do not included also the biomass analyzed in the fish group rearing at colder temperature at the different stages of rearing?
Lines 247-256. All this seems to be more introduction than discussion.
Author Response

(The authors gave the same response as above.)

Round 2
Reviewer 3 Report
The authors have largely addressed the detailed questions in my previous review. Specifically, the description of the compensatory growth is much more explained in this second round. I think this is a commendable advancement, and therefore I recommend publication.
It is OK that authors should keep table 1 in the manuscript.